

# Indigo alleviates psoriasis through the AhR/NF-κB signaling pathway: an *in vitro* and *in vivo* study

Yu Lin[1,2,*], Lihong Yang[3,4,*], Dongxiang Wang[1], Haiqing Lei[1], Yuelin Zhang[1], Wen Sun[5] and Jing Liu[3,4]

[1] Department of Dermatology, the First Clinical Medical School of Guangzhou University of Chinese Medicine, Guangzhou, China
[2] Department of Dermatology, the First Affliated Hospital of Jinan University, Guangzhou, China
[3] Department of Dermatology, the First Affiliated Hospital of Guangzhou University of Chinese Medicine, Guangzhou, China
[4] Guangdong Clinical Research Academy of Chinese Medicine, Guangzhou, China
[5] Department of Dermatology, Jingmen Central Hospital, Jingmen, China
* These authors contributed equally to this work.

Corresponding authors
Wen Sun, MDSunny@yeah.net
Jing Liu, liujing1350@gzucm.edu.cn

## ABSTRACT

**Background:** Psoriasis is a chronic inflammatory skin disease. A strong association between the AhR/ NFκB axis and the inflammatory response in psoriasis. Indigo (IDG) has demonstrated significant anti-inflammatory properties. This study aimed to assess the anti-psoriatic efficacy of IDG while investigating the underlying mechanisms involved.

**Methods:** In the *in vitro* experiments, cell viability was assessed using the CCK-8. qRT-PCR was employed to measure the mRNA levels of NF-κB, TNF-α, IL-1β, AhR, and CYP1A1. Western blotting was conducted to examine alterations in cytoplasmic and nuclear AhR protein levels. Additionally, an IDG nanoemulsion (NE) cream was prepared for the *in vivo* experiments. A psoriasis-like skin lesion mice model was induced using IMQ (62.5 mg/day for 7 days). The severity of psoriasis was evaluated using PASI, and skin lesions were scored while epidermal thickness was assessed *via* HE staining. The expression of inflammatory markers, including IL-6, IL-13, IL-17A, MCP-1, and TNF-α, was detected in skin lesions using Luminex. The levels of CYP1A1, p65, and p-p65 proteins were determined by Western blotting.

**Results:** LPS stimulation significantly elevated TNF-α, IL-6, and NF-κB mRNA levels, which were notably reduced by IDG treatment. Additionally, IDG significantly enhanced the expression of AhR and CYP1A1 mRNA. Further investigation revealed that IDG facilitated AhR translocation from the cytoplasm to the nucleus. In the IMQ-induced psoriasis-like mouse model, IDG NE substantially ameliorated the severity of skin lesions. Moreover, IDG NE treatment reduced the upregulation of inflammatory cytokines such as IL-6, IL-17A, MCP-1, and TNF-α in IMQ-induced skin lesions. It was also observed that IDG NE treatment increased CYP1A1 protein expression while inhibiting p65 and p-p65 protein expression.

**Conclusion:** IDG emerges as a promising treatment for psoriasis, demonstrating effective therapeutic outcomes. Its mechanism of action is likely linked to the modulation of the AhR/NFκB signaling pathway.

# INTRODUCTION

Psoriasis is an immune-mediated, chronic, and relapsing inflammatory skin disorder. Often associated with joint damage, psoriasis is also linked to a higher risk of metabolic syndrome, cardiovascular disease, and diabetes mellitus, which significantly impairs patients' quality of life and mental well-being (*Armstrong & Read, 2020*). The most prevalent form, chronic plaque psoriasis or psoriasis vulgaris, arises from genetic predispositions, particularly the HLA-C*06:02 risk allele, alongside environmental triggers such as streptococcal infections, stress, smoking, obesity, and alcohol consumption (*Griffiths et al., 2021*). Globally, psoriasis affects approximately 125 million people, with a prevalence of 0.1–3%, and its incidence continues to rise, leading to an increasing demand for effective treatments (*Parisi et al., 2020*).

Treatment options for psoriasis encompass topical, systemic, and alternative therapies. Mild psoriasis is generally managed with topical agents, including glucocorticoids, vitamin D3 analogs, and calcineurin inhibitors. For moderate to severe cases, systemic treatments such as methotrexate (MTX), cyclosporine, retinoids, and biologics are employed. Despite the effectiveness of these treatments in alleviating common psoriatic lesions, they present challenges, including high costs, numerous side effects, and limited long-term efficacy (*Benezeder & Wolf, 2019*). Consequently, the pursuit of long-term treatment options for psoriasis remains a significant focus within the global medical community.

Research has demonstrated that abnormal interactions between keratinocytes (KCs) and the immune system are pivotal in the pathogenesis of psoriasis, with the aryl hydrocarbon receptor (AhR) and nuclear factor (NF)-κB serving as critical mediators of these interactions. AhR, a ligand-activated cytoplasmic transcription factor, plays a significant role in the treatment of psoriasis (*Colonna, 2014*; *Di Meglio et al., 2014*). Notably, Tapinarof, a small molecule non-steroidal AhR agonist, has proven effective in treating plaque psoriasis across multiple clinical trials, exhibiting safety and tolerability with minimal adverse effects (*Furue, Hashimoto-Hachiya & Tsuji, 2019*; *Robbins et al., 2019*; *Merk, 2019*). NF-κB, a gene transcription regulator, orchestrates inflammatory responses and other complex biological processes, contributing to immunoinflammation, cell proliferation, differentiation, and apoptosis. Numerous studies have shown that NF-κB is involved in regulating the secretion of psoriatic inflammatory cytokines, including interleukin (IL)-1β, IL-17, IL-22, IL-12, and tumor necrosis factor (TNF)-α (*Li et al., 2011*; *Vinuales et al., 2013*; *Giuliani et al., 2001*). Moreover, drugs currently approved for psoriasis treatment, such as TNF-α antagonists, IL-12/IL-17 inhibitors, and glucocorticoids, exert inhibitory effects on the NF-κB signaling pathway (*Vegfors et al., 2012*; *Uluckan & Wagner, 2016*). Recent studies have also uncovered that natural product monomers, such as D-sorbitol, goldenseal isoflavin, and *Rhodiola rosea* glycosides, can alleviate psoriasis-like skin lesions in mice by modulating the NF-κB signaling pathway (*Kim & Park, 2012*; *Wang et al., 2019*; *Xu et al., 2019*). The involvement of both the AhR and NF-κB pathways in psoriasis highlights their essential role in disease progression. The

relationship between these two pathways is a growing area of interest. Recent findings suggest that the AhR pathway can regulate NF-κB, thereby inhibiting inflammation. In a study on vitamin D3 treatment for periodontitis in an animal model, *Li et al. (2019)* discovered that VD3 could inhibit periodontitis by enhancing AhR activation, which in turn blocked NF-κB binding sites and masked its transcriptional activity, leading to reduced expression of downstream inflammatory genes. Although research on the interplay between AhR and NF-κB signaling in psoriasis is still limited, both pathways have demonstrated significant potential in suppressing psoriasis-related inflammation. The intricate relationship between these pathways could become a crucial target for psoriasis treatment, driving the search for natural products that can modulate both signaling cascades and capturing the growing interest of scholars in this field.

Indigo (IDG), an indole compound, has been recognized for its antioxidant, antibacterial, anti-inflammatory, and antiproliferative properties, making it a potential treatment for various inflammatory conditions (*Zhao et al., 2017*; *Andreazza et al., 2015*; *Xie et al., 2004*; *Kawai et al., 2017*). Research by *Zhao et al. (2017)* demonstrated that IDG effectively scavenges and reduces superoxide anion radicals and 1,1-diphenyl-2-picrylhydrazyl radicals, suggesting its use as a natural antioxidant in the pharmaceutical industry. Additionally, IDG's inhibition of downstream protein expression has been linked to the suppression of NF-κB activity, which subsequently reduces the expression of downstream inflammatory factors. This process decreases the release of inflammatory mediators at psoriasis lesions, thereby alleviating symptoms (*Zhang et al., 2022*). However, studies specifically focusing on the therapeutic effects of IDG on psoriasis are limited, and its poor water solubility further restricts its use as a topical treatment. To address these challenges, our team developed a nanoemulsion (NE) system to enhance IDG's skin penetration. The findings revealed that topical application of IDG NE effectively inhibited imiquimod (IMQ)-induced psoriasis-like skin lesions in mice. This therapeutic effect may be attributed to the activation of the AhR signaling pathway and the inhibition of the NF-κB signaling pathway. Furthermore, this study also provided insights into the possible mechanisms underlying IDG's treatment of psoriasis in HaCaT cells *in vitro*.

## MATERIALS AND METHODS

### Reagents and antibodies

IDG (Pubchem CID: 10215, 99% purity, verified by high-performance liquid chromatography) was procured from Chengdu Ruifenshi Biotechnology Co., Ltd. (Chengdu, Sichuan, China), with its chemical structure depicted in Fig. 1. Additional materials included glycerol, polysorbate-80, caprylic/capric triglyceride, p-hydroxyacetophenone, 1,2-hexanediol, and sodium polyacryloyldimethyl taurate, all sourced from Shanghai Maclean's Biochemical Science and Technology Co., Ltd. (Shanghai, China). The imiquimod 5% cream was obtained from Sichuan Mingxin Pharmaceutical Co., Ltd. (Chengdu, China) (Approval No.: State Drug Permit H20030128), and the BCA Protein Assay Kit and MOUSE CYTOKINE/CHEMOKINE MAGNETIC BEAD PANEL were purchased from Poly Research Biotechnology Co., Ltd. (Warrington, PA, USA).

**Figure 1 IDG chemical structure diagram.**

## Preparation of IDG NE

To prepare the IDG NE, 5% glycerol and 0.1% IDG were individually mixed and pre-dispersed. Meanwhile, 5% glycerol, 0.3% p-hydroxyacetophenone, and 0.5% 1,2-hexanediol were dissolved by stirring in a water bath heated to 80 °C. The two mixtures were then combined, followed by the addition of 0.8% polyacryloyldimethyltaurine and 83.8% purified water, with continued stirring until homogeneously dispersed. Subsequently, 3% polysorbate-80 and 1.5% caprylic/capric triglyceride were incorporated under constant heating at 60 °C with stirring to ensure thorough mixing. The mixture was then subjected to high-speed shearing using a homogenizer for 5–8 min, resulting in the final 0.1% IDG NE. The blank nanoemulsion and 1% IDG NE were prepared using a similar procedure.

## Animals

SPF-grade BALB/c mice, male, aged 5–6 weeks and weighing 15–17 g, were sourced from the Guangdong Laboratory Animal Centre (Laboratory Animal Production License No.: SCXK (Guangdong) 2022-0002; Laboratory Animal Qualification Certificate No.: 44007200117320). The mice were housed in the animal facility of Guangzhou University of Traditional Chinese Medicine, maintained under a 12-h light/12-h dark cycle at a controlled room temperature of 22–25 °C with a relative humidity of 55–70%. Six mice were housed per cage with free access to standard food and water. All experimental protocols were approved by the Laboratory Animal Ethics Committee of Guangzhou University of Traditional Chinese Medicine (No. 20221209004).

Throughout the study, continuous monitoring of the mice was conducted to assess health status, behavioral changes, and any signs of distress. This included regular body weight measurements, evaluations of coat condition, checks for skin damage, and inspections for eye and nose discharge. Additionally, food and water intake were recorded, social behaviors were observed, and any instances of abnormal grunting, unnatural body posture, or excessive licking or scratching were documented.

Euthanasia was carried out when mice exhibited severe illness, irrecoverable health issues, significant behavioral abnormalities, or upon reaching the experimental endpoint. The euthanasia procedure involved administering an overdose of anesthetic to ensure a rapid and painless death.

## IMQ-induced psoriasis-like mouse model

Following a 1-week acclimatization period, the mice were shaved on their backs to create a hair-free area of approximately 3 cm × 4 cm. Thirty-six male BALB/c mice were then randomly divided into six groups using a random number table: a blank control group, a model group, a dexamethasone (DXM) control group, a blank NE group, a 0.1% IDG NE group, and a 1% IDG NE group, with six mice per group. In the blank control group, no treatment was applied to the shaved backs of the mice. In the model group, after shaving, 62.5 mg of 5% IMQ cream was applied once daily to the back for psoriasis modeling. For the DXM control group, in addition to the daily application of 5% IMQ cream, compounded DXM acetate cream was applied once in the morning and once in the evening at a dose of approximately 70 mg each time. The remaining groups, including the blank NE group, 0.1% IDG NE group, and 1% IDG NE group, received 70 mg of their respective treatments (blank NE, 0.1% IDG NE, and 1% IDG NE) once in the morning and once in the evening, in conjunction with the daily IMQ application.

## Scoring system

The progression of skin lesions on the backs of the mice in each group was observed daily, with photographs taken on days 1, 4, and 7 for comparative analysis. The severity of the lesions was assessed using the Psoriasis Area and Severity Index (PASI) scale, focusing on erythema, desquamation, and infiltration. Each characteristic was scored on a scale from 0 to 4, where 0 indicated no lesions, 1 mild, 2 moderate, 3 severe, and 4 extremely severe. The total score was calculated by summing the scores for erythema, desquamation, and infiltration, resulting in a possible range of 0–12 points.

Mice in each group were weighed on days 1, 4, and 7, with weights recorded accordingly. On the seventh day of treatment, after the mice were sacrificed, their spleens were weighed to calculate the spleen index, which was determined by the ratio of spleen weight to body weight (spleen index = spleen weight/body weight).

## Histopathology

All mice were euthanized on the seventh day of treatment through intraperitoneal injection of 150 mg/kg sodium pentobarbital. Criteria for early euthanasia were established to prioritize animal welfare, including signs of severe distress, weight loss exceeding 20% of body weight, or other significant health concerns. During the experiment, no mice met these criteria for early euthanasia. Post-euthanasia, skin samples from the dorsal lesions were collected and fixed in formalin. The skin tissues were then dehydrated using a gradient alcohol process, embedded in paraffin, sectioned, and stained with hematoxylin-eosin (HE). Pathological analysis was conducted by randomly selecting five fields of view from individual tissue slices under a 200X light microscope, with pathological images captured for further analysis. In cases where mice exhibited signs of impending mortality during the experiment, euthanasia was administered immediately. At the conclusion of the experiment, all surviving mice were humanely euthanized.

## Luminex liquid chip detection of cell cytokines

Luminex liquid microarray technology was employed to detect cytokines in both serum and skin tissue, specifically IL-6, IL-13, IL-17A, TNF-$\alpha$, and MCP-1, with each sample subjected to three experimental repetitions. Total proteins were extracted using RIPA buffer (CST, Boston, MA, USA) and then centrifuged at 12,000 rpm for 5 min at 4 °C. Protein concentrations were determined using a BCA protein assay kit (Merck, Darmstadt, Germany). For the assay, 25 µL of each sample was added to 96-well microtiter plates. Beads were prepared by extracting 60 µL from each tube into a mixing bottle, diluting to 3 mL with bead diluent, and then adding 50 µL to each well. The plates were incubated by shaking them at room temperature for 2 h, followed by two washes with a washing buffer. Subsequently, 25 µL of detection antibody was added to each well and incubated with shaking for 1 h at room temperature. This was followed by the addition of 25 µL of SAPE, with further incubation and shaking for 30 min at room temperature. The plates were washed twice, and 150 µL of sheath solution was added to each well and shaken for 5 min at room temperature in the dark. The cytokine content was measured using the MILLIPLEX® MAGPIX system, and data analysis was performed using MILLIPLEX Analyst V5.1 software.

## Cell culture and drug treatment

The human immortalized keratinocyte cell line (HaCaT) was obtained from the Chinese Academy of Science Center for Excellence in Molecular Cell Science and cultured in DMEM supplemented with 10% heat-inactivated FBS. Cells were maintained at 37 °C in a sterile incubator with 5% $CO_2$. HaCaT cells, which are commonly used to model oxidative stress and inflammatory injury, were stimulated with 1 µg/mL lipopolysaccharide (LPS) for 4 h to establish a psoriasis-like cell model. Following this induction, the cells were treated with 48 µM of IDG for an additional 24 h.

## Cell counting kit-8 (CCK-8) assay

The IC50, or the concentration of an inhibitor required to achieve 50% inhibition, was determined in this study. After culturing the cells for 24 h, they were treated with varying concentrations of IDG for an additional 24 h. Cell viability was then assessed using the CCK-8 kit (Japan Tongren Company, Dojindo, Japan). Following the addition of the CCK-8 reaction solution, the cells were incubated in the dark for 3 h. Absorbance was measured at a wavelength of 450 nm using a full-wavelength enzyme-linked immunosorbent assay reader (Thermo Fisher Scientific, Waltham, MA, USA), and cell survival rates were calculated according to the manufacturer's formula.

## Real-time polymerase chain reaction (RT-PCR)

RT-qPCR was employed to detect the expression levels of NF-κB, TNF-$\alpha$, IL-1$\beta$, AhR, and CYP1A1 genes in HaCaT cells. The experiment included three groups: a blank control group (CON), a model group (LPS), and a drug treatment group (IDG). HaCaT cells in the logarithmic growth phase were seeded in six-well plates at a density of $2 \times 10^6$ cells per well. After overnight incubation to achieve complete adherence, the drug group was

treated with 48 µM IDG, and 24 h later, both the drug and model groups were stimulated with 1 µg/mL LPS. Following a 4-h incubation, samples were collected from the cell culture box. Total RNA was extracted on ice using TRIzol® Reagent (#15596026; Thermo Fisher, China, Hong Kong). The concentration of RNA samples was measured using an ultra-micro spectrophotometer, with RNA purity assessed by the A260/A280 ratio, which ideally should be around 2.0; deviations from this value suggest potential contamination during the experimental process. The RNA concentration was diluted to 1 µg/µl, and genomic DNA was digested and reverse-transcribed into cDNA using the Evo M-MLV RT Mix Kit with gDNA Clean for qPCR II (#AG11711; Accurate Biotechnology, Shenzhen, China). The reaction mixture, including 1 µl RNA, gDNA Clean Reagent, 5X gDNA Clean Buffer, and other components, was incubated at 42 °C for 5 min to remove gDNA, followed by reverse transcription at 37 °C for 15 min and at 85 °C for 5 s. The resulting cDNA was stored at −80 °C. Subsequently, the qRT-PCR reaction was set up according to the instructions of the SYBR® Green Premix Pro Taq HS qPCR Kit (#AG11701; Accurate Biotechnology, Shenzhen, China). The 20 µl reaction mixture consisted of 2 × SYBR® Green Pro Taq HS Premix, 0.2 µM each of upstream and downstream primers, cDNA, and RNase-free water. The PCR amplification was performed using the LABI 7500 system (Applied Biosystems) under the following conditions: initial denaturation at 95 °C for 30 s, followed by 40 cycles of 95 °C for 5 s and 60 °C for 30 s. The melt curve analysis showed a single peak, and no amplification was observed in the No Template Control (NTC), confirming the specificity of the reaction. Primers for NF-κB, TNF-α, IL-1β, AhR, and CYP1A1 were synthesized by Shanghai Jieri Bioengineering Co., Ltd. (Table 1). The relative expression levels of target genes were calculated using the $2^{(-\Delta\Delta Ct)}$ method, with GAPDH serving as the internal control. The PCR efficiency was validated, with slopes ranging from −3.6 to 3.1 and an $R^2$ value of at least 0.98. A limit of detection (LOD) of 2.5 was established within a 95% confidence interval. Data analysis involved the exclusion of technical replicates that deviated significantly from the other two within each triplicate set. Each group comprised eight biological replicates, with each sample subjected to three technical replicates to ensure robust and reliable results.

## Western blotting analyses

After extracting and isolating cytosolic and nuclear proteins using the Total and Nuclear Protein Extraction Kit (Solebo), protein concentrations were determined using the BCA Protein Assay Kit (Beyotime). Proteins were denatured by boiling at 100 °C for 5 min. Protein samples (25 µg per sample) were then separated by electrophoresis on an SDS-PAGE gel (Beyotime). The separated proteins were subsequently transferred to PVDF membranes, which were blocked for 1 h at room temperature in Tris-buffered saline with 0.1% Tween 20 (TBST) containing 5% skimmed milk powder. The membranes were incubated overnight at 4 °C with the respective primary antibodies: β-actin (1:10,000, affinity) and AHR (1:1,000; Affinity). Following this, the membranes were washed four times with TBST (5 min each) and incubated with horseradish peroxidase (HRP)-coupled secondary antibodies: goat anti-rabbit IgG (1:2,000; Affinity) and goat anti-mouse IgG for 2 h at room temperature. The membranes were then washed four times with TBST (5 min

**Table 1 Primer sequences of the genes.**

| Gene | Primer sequences (5′-3′) |
| --- | --- |
| AhR | F: ACAACCGATGGACTTGGGTC |
|  | R: TGGCAGGAAAAGGGTTGGTT |
| CYP1A1 | F: AATTTCGGGGAGGTGGTTGG |
|  | R: AGGCATTCAGGGAAGGGTTG |
| NF-κB | F: AACAGAGAGGATTTCGTTTCCG |
|  | R: TTTGACCTGAGGGTAAGACTTCT |
| IL-1β | F: ATGATGGCTTATTACAGTGGCAA |
|  | R: GTCGGAGATTCGTAGCTGGA |
| TNF-α | F: GAGGCCAAGCCCTGGTATG |
|  | R: CGGGCCGATTGATCTCAGC |
| GAPDH | F: GAAAGCCTGCCGGTGACTAA |
|  | R: GCATCACCCGGAGGAGAAAT |

each). Protein detection was conducted using a Bio-Rad gel imaging system after applying the ECL luminescent solution (Beyotime) to the membranes. The grey values of the protein bands were analyzed using ImageJ software.

## Statistical analysis

Data are presented as the mean ± standard error of the mean (SEM) from at least three independent experiments. The Kolmogorov-Smirnov (K-S) test was used to assess the normality of the data distribution. For data that followed a normal distribution, *one-way ANOVA* was performed; otherwise, the non-parametric "Independent-Samples Kruskal-Wallis Test" was used. A *P-value of <0.05* was considered statistically significant. Statistical analyses were conducted using SPSS 26.0 (SPSS, Inc., Chicago, IL, USA), and figures were generated using GraphPad Prism 9.5 (GraphPad Software, Inc., La Jolla, CA, USA).

## RESULTS

### LPS stimulation promotes the expression of inflammatory factors in HaCaT keratinocytes

A psoriasis-like cell model was established by stimulating HaCaT keratinocytes with LPS, aiming to replicate key features of psoriasis *in vitro* (*Takuathung et al., 2021*; *Thatikonda et al., 2020*; *Lin et al., 2023*; *Hong et al., 2022*). The qRT-PCR results demonstrated that 1 µg/mL LPS stimulation of HaCaT cells for 4 h significantly upregulated the mRNA levels of TNF-α, IL-6, and NF-κB (Fig. 2A). This finding confirms that LPS at a concentration of 1 µg/mL can effectively induce HaCaT keratinocytes to mimic certain characteristics of psoriatic keratinocytes. Consequently, LPS (1 µg/mL) was used to establish an *in vitro* psoriasis-like cell model using HaCaT keratinocytes.

## IDG inhibits inflammatory responses in LPS-stimulated keratinocytes

The IC50 of IDG on HaCaT cells was determined using the CCK-8 assay. Following a 24-h treatment with varying concentrations of IDG, the IC50 was calculated to be 118.7 μmol/L (*Smolinska et al., 2018*; *Bai et al., 2023*) (Fig. 2B). For subsequent experiments, an approximation of 40% of the IC50, equivalent to 48 μmol/L, was selected as the experimental concentration. The impact of IDG on LPS-stimulated HaCaT keratinocytes was evaluated by qPCR. The results indicated that LPS stimulation markedly elevated the mRNA levels of NF-κB, IL-1β, and TNF-α, but these levels were significantly reduced in the presence of IDG (Figs. 2C–2E).

## IDG activates AhR signaling in LPS-induced HaCaT keratinocytes

The activation of the AhR has been well-established as an effective treatment strategy for psoriasis. AhR is a ligand-dependent transcription factor, and upon ligand binding, its nuclear translocation site becomes exposed, facilitating the translocation of AhR into the nucleus. Once in the nucleus, AhR forms a heterodimer with the aryl hydrocarbon receptor nuclear translocator (ARNT), which subsequently regulates downstream genes such as CYP1A1, playing a critical role in controlling cellular inflammation (*Di Meglio, Perera & Nestle, 2011*). To explore whether IDG mitigates the inflammatory response in HaCaT keratinocytes through the activation of the AhR pathway, we measured the mRNA levels of AhR and CYP1A1 using qPCR. The results showed a significant increase in AhR and CYP1A1 mRNA levels following IDG treatment (Figs. 3A and 3B). Additionally, to further elucidate the cellular localization of AhR, Western blotting analysis was performed. The results indicated that under normal conditions, AhR, a constitutively expressed molecule, is predominantly located in the cytoplasm. However, upon treatment with IDG, increased nuclear translocation of AhR was observed compared to the untreated condition (Figs. 3C–3E).

## IDG NE ameliorates skin lesions in IMQ-induced psoriasis mice

To assess the therapeutic effect of IDG NE on an IMQ-induced psoriasis-like mouse model, topical treatments with IDG NE were administered. Initial observations of dorsal skin lesions and PASI scores following IMQ administration displayed characteristic psoriasis-like manifestations. Both IDG NE and DXM treatments significantly alleviated symptoms such as erythema, scaling, and skin thickening, resulting in a substantial reduction in PASI scores (Figs. 4A and 4C). Patients with psoriasis often exhibit splenomegaly, which may be indicative of the immune system's chronic inflammatory response (*Balato et al., 2015*). Therefore, the spleen index was employed as a key indicator in this study. Findings revealed a significant increase in spleen index among IMQ-treated mice compared to the control group, while IDG NE treatment significantly reduced the spleen index in these mice (Figs. 4D–4F). H&E staining further demonstrated that IMQ induction caused distinct pathological changes in psoriasis-like lesions, characterized by excessive or incomplete epidermal keratinization, thickening of the acanthosis cell layer, and pronounced downward proliferation of epidermal ridges. Remarkably, IDG NE treatment mitigated these histological abnormalities (Figs. 4B and 4G).

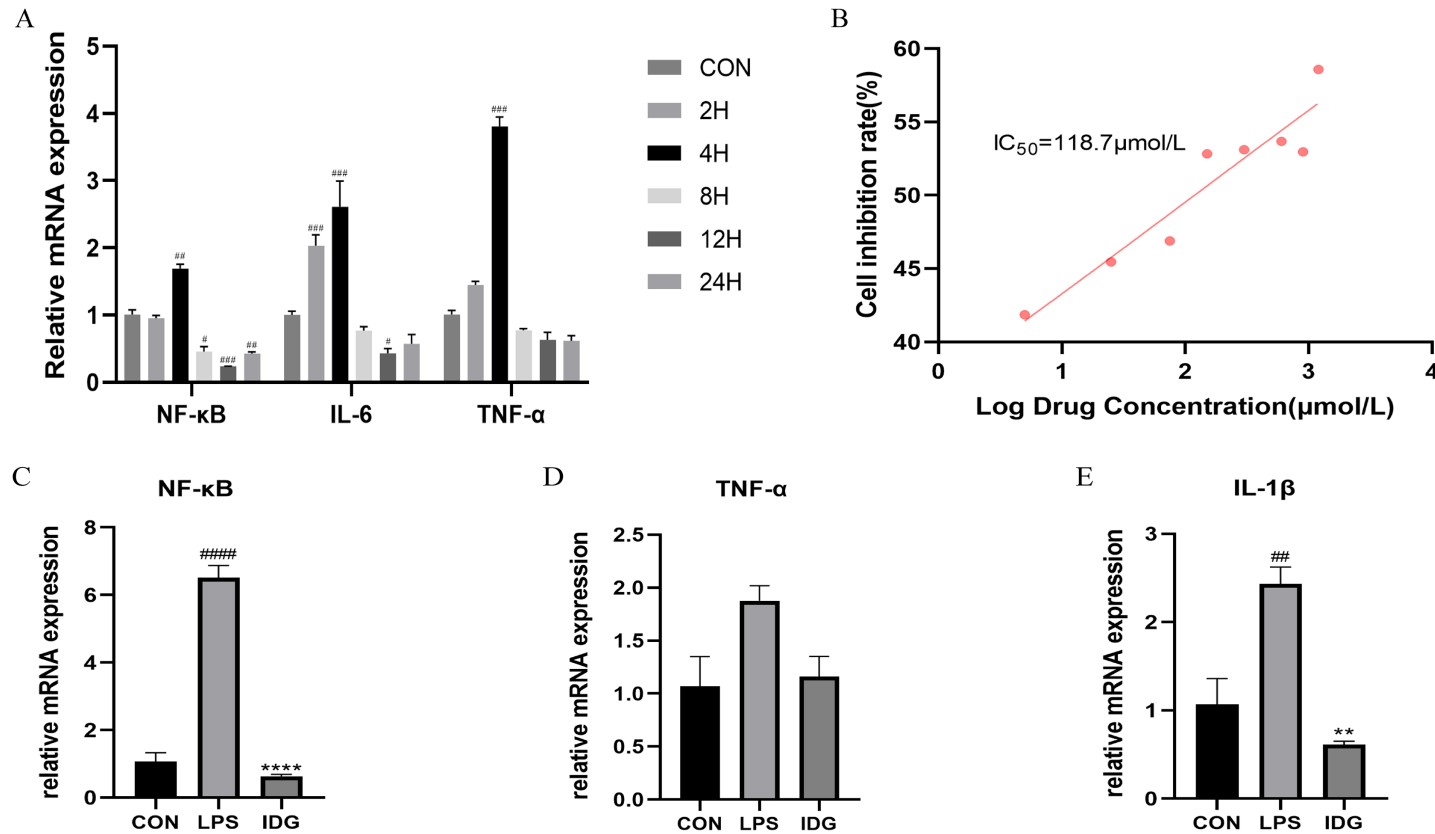

**Figure 2 IDG inhibits inflammatory responses in LPS-stimulated keratinocytes.** (A) The mRNA expressions of TNF-α, IL-6 and NF-κB after different times of LPS intervention in HaCaT cells. (B) Inhibition rate of HaCaT cells by different concentrations of IDG. (C–E) Effect of IDG on mRNA expressions of NF-κB, TNF-α, IL-1β in HaCaT cell psoriasis-like model. $^{####}P < 0.0001$, $^{###}P < 0.001$, $^{##}P < 0.01$, $^{#}P < 0.05$ *vs* CON; $^{****}P < 0.0001$, $^{**}P < 0.01$ *vs* LPS.

Additionally, the IDG NE utilized in the study demonstrated a particle size of 180.7 ± 3.5 nm and a polydispersity index of 0.28 ± 0.01 ($n = 3$), as determined by laser particle size analysis. Stability tests confirmed the formulation's excellent long-term stability.

## IDG NE suppresses IMQ-triggered skin inflammatory cytokines and chemokines

To assess the impact of IDG NE on cytokine levels in skin tissue and serum in IMQ-induced mice, Luminex liquid microarrays were employed to measure inflammatory cytokines. Results indicated that IDG and DXM significantly inhibited the expression of IL-6, IL-17A, MCP-1, and TNF-α in psoriasis-like lesions, with all findings reaching statistical significance, while the effect on IL-13 was not statistically significant (Table 2). In the serum of IMQ-induced mice, levels of IL-6, IL-13, IL-17A, MCP-1, and TNF-α were significantly elevated in the IMQ group but were notably reduced in both the IDG and DXM groups, with all reductions being statistically significant (Table 3).

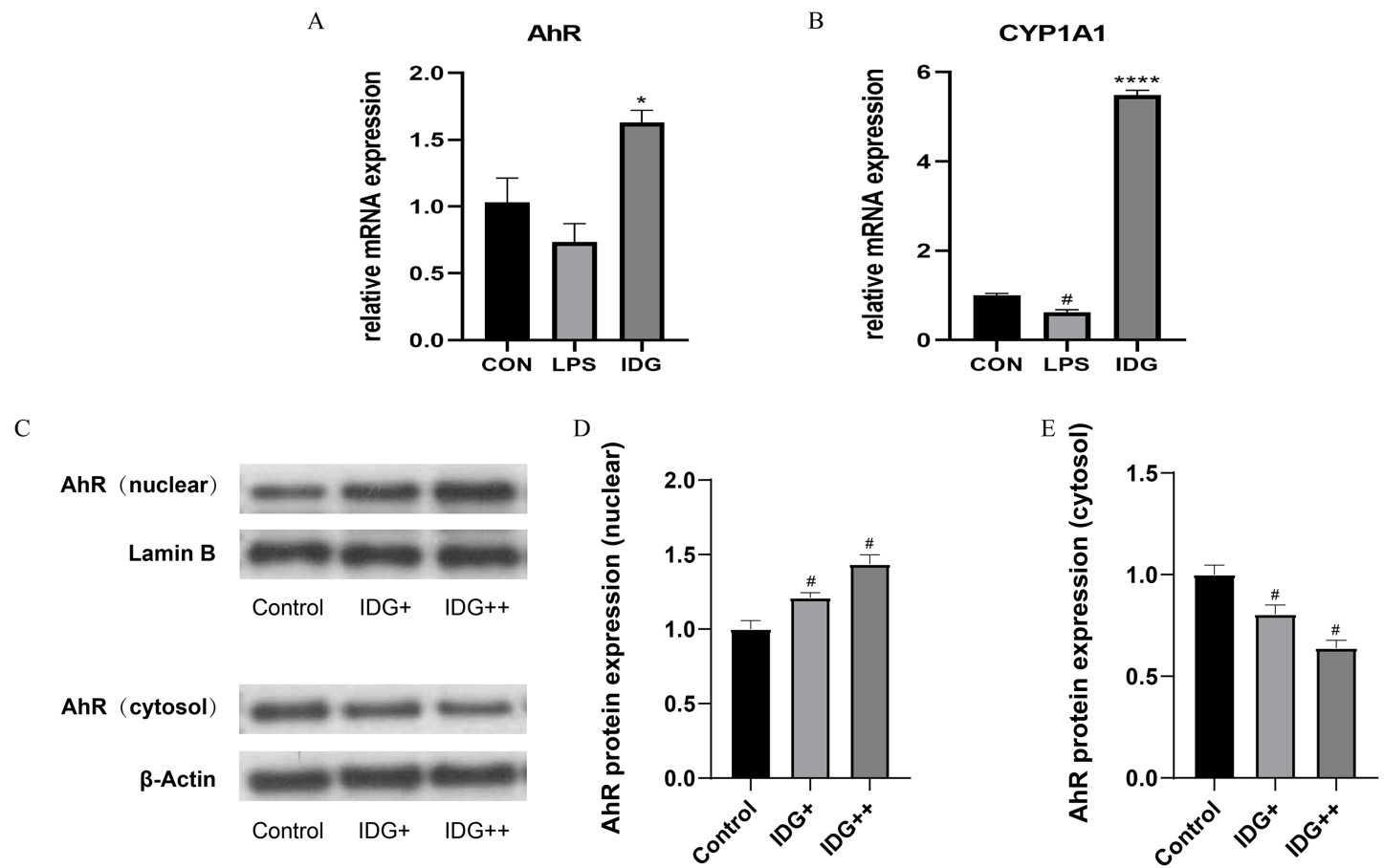

**Figure 3 IDG could activate AhR signaling in LPS-induced HaCaT keratinocytes.** (A, B) Effect of IDG on mRNA expressions of AhR and CYP1A1 in HaCaT cell psoriasis-like model. (C–E) Nuclear and cytoplasmic proteins of AhR and were detected by using western blotting (WB), respectively. $^{\#}P < 0.05$ *vs* CON; $^{****}P < 0.0001$, $^{*}P < 0.05$ *vs* LPS.

## IDG NE reduces IMQ-induced psoriasis-like skin inflammation *via* activating AhR signaling

Further investigation into whether IDG NE mitigates IMQ-induced psoriasis-like skin inflammation through the activation of AhR signaling was conducted *via* Western blotting analysis to detect CYP1A1 protein expression in skin lesion tissues. Results revealed a significant reduction in CYP1A1 expression within the model group. In contrast, both IDG NE and DXM treatments significantly enhanced CYP1A1 expression, with IDG NE showing a more pronounced effect than DXM, suggesting that IDG NE facilitates AhR activation (Figs. 5A and 5B).

## IDG NE inhibits NFκB-mediated inflammatory responses

NF-κB, a family of pleiotropic transcription factors, is ubiquitously expressed across various cell types and comprises five members: NF-κB1 (p105/p50), NF-κB2 (p100/p52), Rel-A (p65), Rel-B, and c-Rel. Activation of the classical NF-κB pathway primarily involves p50 and p65. Under resting conditions, NF-κB forms a complex with its inhibitor

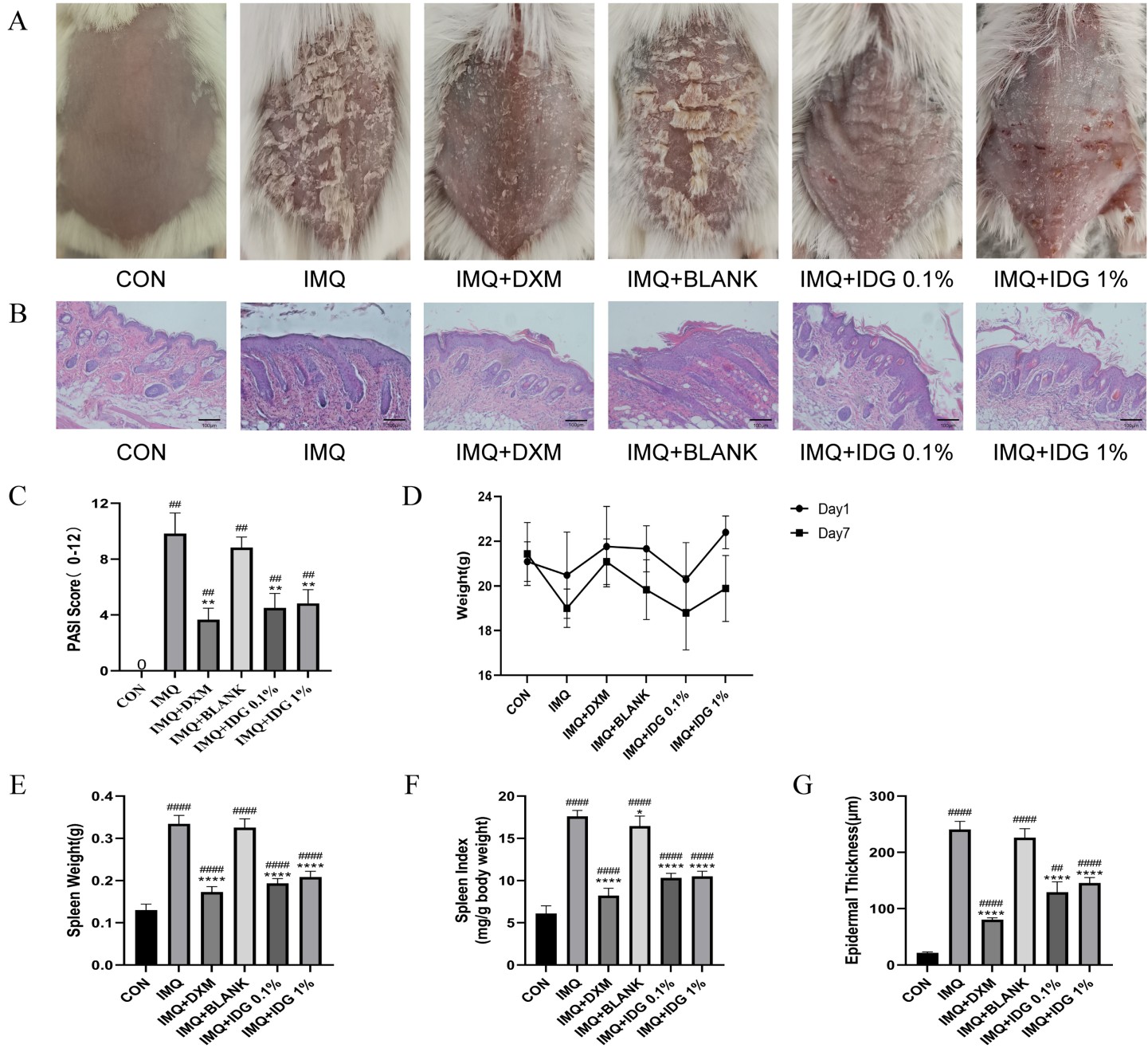

**Figure 4 IDG NE ameliorates skin lesion in IMQ-induced psoriasis mice.** (A) The macroscopic appearance of the skin tissue; (B) Histological evaluation of the back skin of IMQ-induced psoriasis-like mice. (H&E staining; magnification 100×). (C) the PASI scores of the skin tissue. (D) The weight of mice. (E) The spleen weight of mice. (F) The spleen index of mice. (G) Skin thickness. $^{####}P < 0.0001$, $^{##}P < 0.01$ $vs$ CON, $^{****}P < 0.0001$, $^{**}P < 0.01$, $^{*}P < 0.05$ $vs$ IMQ.

IκB in the cytoplasm. Upon degradation of IκB, p65 is phosphorylated into p-p65, allowing its release and translocation into the nucleus, where it initiates the transcription of target genes (*Giuliani et al., 2001*). In this study, the expression levels of NF-κB p65 and p-NF-κB p65 were examined in the skin lesions of each group. IMQ pretreatment significantly

**Table 2 Cytokines in psoriasis-like lesions (mean ± SEM, $n = 6$).**

| Cytokine (pg/mL) | CON | IMQ | IMQ+DXM | IMQ+IDG 0.1% | IMQ+IDG 1% |
|---|---|---|---|---|---|
| IL-6 | 2.66 ± 0.66 | 17.54 ± 4.37[###] | 10.47 ± 5.01 | 7.45 ± 1.44[**] | 11.09 ± 4.75 |
| IL-13 | 8.79 ± 2.51 | 18.88 ± 11.82 | 11.14 ± 9.74 | 4.58 ± 3.98[**] | 4.18 ± 3.97[**] |
| IL-17A | 0.73 ± 0.36 | 2.12 ± 0.98[##] | 0.63 ± 0.34[**] | 0.75 ± 0.31[**] | 0.70 ± 0.19[**] |
| MCP-1 | 2.32 ± 0.93 | 6.02 ± 2.43[#] | 3.56 ± 1.87 | 1.33 ± 0.65[***] | 2.74 ± 2.34[*] |
| TNF-α | 1.81 ± 0.82 | 19.16 ± 4.29[##] | 11.74 ± 5.22 | 4.68 ± 2.90[**] | 10.13 ± 3.33 |

Notes:
[###] $P < 0.001$.
[##] $P < 0.01$.
[#] $P < 0.05$ *vs* CON.
[***] $P < 0.001$.
[**] $P < 0.01$.
[*] $P < 0.05$ *vs* IMQ.

**Table 3 Cytokines in serum of psoriasis-like mice (mean ± SEM, $n = 6$).**

| Cytokine (pg/mL) | CON | IMQ | IMQ+DXM | IMQ+IDG 0.1% | IMQ+IDG 1% |
|---|---|---|---|---|---|
| IL-6 | 21.28 ± 14.87 | 46.54 ± 17.32[#] | 14.48 ± 13.78[**] | 31.49 ± 16.80 | 14.24 ± 11.53[**] |
| IL-13 | 37.41 ± 6.44 | 93.29 ± 19.37[##] | 57.81 ± 16.20 | 51.07 ± 16.21 | 47.09 ± 10.56[*] |
| IL-17A | 7.21 ± 3.13 | 16.49 ± 1.19[####] | 3.84 ± 1.42[****] | 10.19 ± 1.51[****] | 7.98 ± 2.05[****] |
| MCP-1 | 8.82 ± 2.26 | 18.36 ± 2.47[#] | 9.75 ± 1.13 | 8.13 ± 1.82[**] | 7.17 ± 2.66[**] |
| TNF-α | 51.40 ± 16.09 | 90.12 ± 32.42[####] | 71.34 ± 10.82 | 68.20 ± 21.59[*] | 62.19 ± 19.14[**] |

Notes:
[####] $P < 0.0001$.
[##] $P < 0.01$.
[#] $P < 0.05$ *vs* CON.
[****] $P < 0.0001$.
[**] $P < 0.01$.
[*] $P < 0.05$ *vs* IMQ.

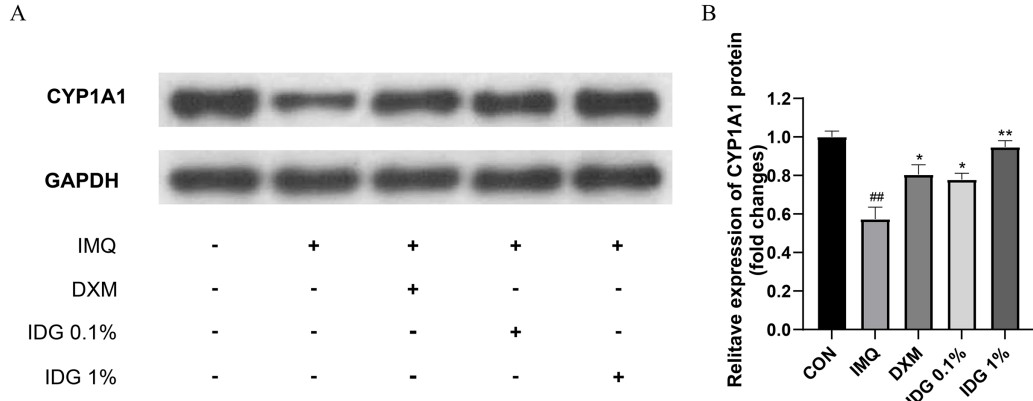

**Figure 5 Indigo could activate AhR signaling in epidermis of psoriasis-like mice.** (A, B) CYP1A1 levels were assessed by western blotting. [##]$P < 0.01$ *vs* CON; [**]$P < 0.01$, [*]$P < 0.05$ *vs* IMQ.

increased the expression of p65 and p-p65, both of which were markedly reduced by IDG NE treatment. Notably, the inhibitory effect of 1% IDG NE was more substantial compared to DXM (Figs. 6A–6D).

## DISCUSSION

Psoriasis is a chronic, immune-mediated skin disease characterized by erythema, infiltration, and desquamation. The cosmetic disfigurement and recurrent nature of psoriasis place significant psychological and economic burdens on patients, severely impacting their quality of life. IDG, a natural indole compound, has demonstrated potent anti-inflammatory properties. This study aimed to elucidate the anti-inflammatory mechanisms of IDG through *in vitro* experiments and to evaluate the therapeutic efficacy of IDG NE in treating imiquimod (IMQ)-induced psoriatic dermatitis in mice.

HaCaT cells, immortalized human keratinocytes derived from normal skin tissues, retain the essential characteristics of human keratinocytes and are frequently utilized in *in vitro* models to study dermatological conditions (*Xiong et al., 2015*; *Hollywood et al., 2015*; *Varma et al., 2017*). LPS, an antigenic complex from the outer membrane of Gram-negative bacteria, is a potent immune system activator. Through binding to TLR4, LPS triggers signaling pathways such as NF-κB and MAPKs, leading to the upregulation of pro-inflammatory cytokines, including IL-6, IL-1β, and TNF-α (*Lu, Yeh & Ohashi, 2008*). The induction of pro-inflammatory cytokines and keratinocyte proliferation by LPS in HaCaT cells mirrors the cellular-level pathological changes observed in psoriasis, making LPS-induced HaCaT cells a well-established *in vitro* psoriasis model. In this study, HaCaT cells were treated with 1 μg/mL LPS (*Smolinska et al., 2018*) for 4 h to create an inflammatory model representative of psoriasis. The significant increase in NF-κB, IL-1β, and TNF-α mRNA levels in the LPS group confirmed the successful construction of the model. Notably, IDG intervention significantly suppressed the expression of these inflammatory markers, indicating that IDG exerts strong anti-inflammatory effects on HaCaT cells *in vitro* and holds promise as a potential therapeutic agent for psoriasis.

IMQ, a small-molecule immunomodulator, is known to induce psoriasis-like skin lesions in mice through mechanisms likely involving inflammatory cytokines such as TNF-α (*Rajan & Langtry, 2006*). The IMQ-induced psoriasis-like mouse model is widely recognized for its ease of use, cost-effectiveness, and ability to replicate topical psoriatic lesions, making it a standard method for psoriasis research. In this study, the IMQ-induced psoriasis-like mouse model was employed to evaluate the therapeutic effects of IDG NE. The findings revealed that IDG intervention significantly reduced the PASI scores of IMQ-induced back skin lesions and alleviated psoriasis-like pathological manifestations, providing preliminary evidence of IDG NE's efficacy in treating psoriasis. Additionally, IDG was shown to inhibit the expression of IL-6, IL-13, IL-17A, TNF-α, and MCP-1, suggesting that IDG may exert its therapeutic effects on psoriasis through its anti-inflammatory properties.

As a ligand-dependent transcription factor, AHR translocates to the nucleus upon ligand binding, where it recognizes promoters containing specific enhancer sequences known as xenobiotic response elements (XREs), which regulate downstream genes such as CYP1A1, CYP1A2, and CYP1B1. Previous research has established that AHR can significantly inhibit T cell infiltration and the migration of inflammatory factors, including interferon-γ, IL-17, and TNF-α (*Furue, Hashimoto-Hachiya & Tsuji, 2019*; *Robbins et al.,*

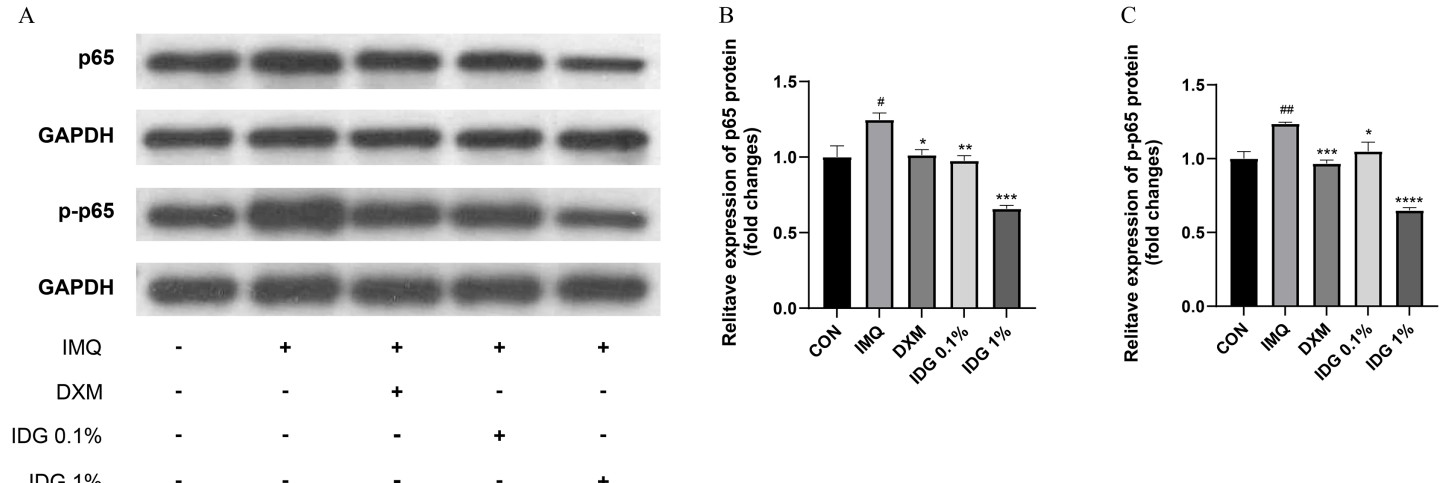

**Figure 6 Indigo inhibited NF-κB signaling in epidermis of psoriasis-like mice.** (A–D) P-p65 and p65 levels were assessed by western blotting. $^\#P < 0.05$, $^{\#\#}P < 0.01$ *vs* CON ;$^{****}P < 0.0001$, $^{***}P < 0.001$, $^{**}P < 0.01$, $^{*}P < 0.05$ *vs* IMQ. 

*2019*; *Merk, 2019*). In 2019, Tapinarof (1% benvitimod cream), a natural AHR agonist, was officially approved by the Chinese government for the treatment of psoriasis following successful clinical trials in China (*Furue, Hashimoto-Hachiya & Tsuji, 2019*). Notably, our experiments identified AHR as a therapeutic target of IDG, both *in vivo* and *ex vivo*, demonstrating that IDG can treat psoriasis by activating AHR. These findings suggest that IDG holds promise as the next natural compound for effectively treating psoriasis.

Additionally, IDG was observed to suppress the phosphorylation of p65, thereby blocking the activation of the NFκB pathway triggered by IMQ and inhibiting the expression of inflammatory cytokines. Given the critical role of the NFκB signaling pathway in regulating the inflammatory response, it is a prominent therapeutic target for inflammatory diseases. Therefore, it is reasonable to postulate that NFκB is a key regulator through which IDG ameliorates IMQ-induced psoriasis-like inflammation in mice. Previous studies have demonstrated that AHR interacts with two key NF-κB subunits, p65 and RelB, and that this interaction plays a pivotal role in regulating inflammation (*Vogel & Matsumura, 2009*). Experiments blocking NF-κB and AHR have shown that the expression of several inflammatory cytokines, including IL-17A, TNF-α, MCP-1, and IL-1β, is inhibited and reduced, suggesting that these cytokines' expression may be partially regulated by the interaction between AHR and NFκB (*Denison & Nagy, 2003*). Furthermore, it has been shown that the activation of AHR by various endogenous ligands inhibits the NF-κB signaling pathway, reducing the expression of IL-1β and IL-6 in conditions such as periodontitis, bronchitis, and colitis (*Cui et al., 2022*; *Li et al., 2019*; *Ovrevik et al., 2014*; *Yu et al., 2018*). Given the critical roles of both AHR and NF-κB signaling pathways in psoriasis, our study demonstrated that IDG acts as a ligand for AHR, activates the AHR signaling pathway, and inhibits the NF-κB signaling pathway. This suggests that IDG may exert its anti-inflammatory effects in psoriasis by modulating the AHR/NF-κB signaling pathway. Therefore, it is plausible to speculate that the anti-inflammatory effects of IDG are linked to the interaction between AHR and NFκB.

In conclusion, both high and low concentrations of IDG NE effectively alleviate IMQ-induced psoriasis-like lesions in mice, with no significant difference observed between the two concentrations. This suggests that IDG NE can achieve therapeutic efficacy even at lower concentrations. Importantly, our study is the first to demonstrate that IDG significantly improves the severity of IMQ-induced psoriasis-like skin lesions in mice and to establish that the mechanism underlying this effect may involve the AHR and NFκB signaling pathways.

However, this study has several limitations due to constraints in time and technology. The research was confined to mouse models and psoriasis cell models, which differ significantly from human patients. Additionally, the exploration of the AhR and NF-κB signaling pathways remains at a preliminary stage. Further investigation is needed to elucidate the specific upstream and downstream target genes, transcription factors, and the crosstalk mechanisms involved. It remains to be determined whether AhR directly regulates NF-κB gene expression or does so indirectly through alternative signaling networks. More in-depth and rigorous studies are required to clarify these specific mechanisms.

## CONCLUSIONS

In summary, IDG effectively inhibited LPS-induced inflammatory responses in HaCaT keratinocytes. In an IMQ-induced psoriasis-like mouse model, IDG NE significantly reduced the severity of skin injury and improved the inflammatory state. The anti-inflammatory effects of IDG observed in both *in vitro* and *in vivo* experiments were linked to the modulation of the AhR/NF-κB signaling pathway. These findings highlight IDG's potential as an anti-psoriasis agent, making it a promising candidate for future drug development.

### Funding

This work was supported by the Guangdong Provincial Department of Education university research project (grant number 2021ZDZX2026), the National Traditional Chinese Medicine Inheritance and Innovation Center research project (grant number 2022QN28), and the Guangdong Provincial Bureau of Traditional Chinese Medicine research project (grant number 20213006). The funders had no role in study design, data collection and analysis, decision to publish, or preparation of the manuscript.

### Grant Disclosures

The following grant information was disclosed by the authors:
Guangdong Provincial Department of Education University: 2021ZDZX2026.
National Traditional Chinese Medicine Inheritance and Innovation Center: 2022QN28.
Guangdong Provincial Bureau of Traditional Chinese Medicine: 20213006.

## Competing Interests

The authors declare that they have no competing interests.

## Author Contributions

- Yu Lin conceived and designed the experiments, performed the experiments, analyzed the data, prepared figures and/or tables, authored or reviewed drafts of the article, and approved the final draft.
- Lihong Yang conceived and designed the experiments, performed the experiments, analyzed the data, prepared figures and/or tables, authored or reviewed drafts of the article, and approved the final draft.
- Dongxiang Wang conceived and designed the experiments, performed the experiments, prepared figures and/or tables, and approved the final draft.
- Haiqing Lei performed the experiments, analyzed the data, authored or reviewed drafts of the article, and approved the final draft.
- Yuelin Zhang analyzed the data, prepared figures and/or tables, and approved the final draft.
- Wen Sun conceived and designed the experiments, authored or reviewed drafts of the article, and approved the final draft.
- Jing Liu conceived and designed the experiments, authored or reviewed drafts of the article, and approved the final draft.

## Animal Ethics

The following information was supplied relating to ethical approvals (*i.e.*, approving body and any reference numbers):

All experimental procedures were approved by the Laboratory Animal Ethics Committee of Guangzhou University of Traditional Chinese Medicine.

## Data Availability

The raw data is available at Figshare: Lin, Yu (2024). raw data.zip. figshare. Journal contribution. https://doi.org/10.6084/m9.figshare.25194218.v2.

## Supplemental Information

Supplemental information for this article can be found online at http://dx.doi.org/10.7717/peerj.18326#supplemental-information.

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
