# Peer review of "Indigo alleviates psoriasis through the AhR/NF-κB signaling pathway: an in vitro and in vivo study"

_PeerJ, doi:10.7717/peerj.18326_

## Round 0.1 · original submission · Minor Revisions

All three reviewers have provided positive feedback on your manuscript, recommending minor revisions. After careful consideration, I concur with their assessment. We look forward to receiving your revised manuscript. Please provide a point-by-point response to the reviewers' comments along with your revisions. Clearly highlight all changes made in the revised version.

Reviewer 1 ·

Basic reporting

no comment

Experimental design

no comment

Validity of the findings

no comment

Additional comments

This study investigated the therapeutic potential of indigo (IDG) nanoemulsion (NE) for psoriasis by observing its effects on imiquimod (IMQ)-induced psoriasis-like skin lesions in mice and exploring its mechanisms in vitro. The results showed that IDG significantly improved pathological scores, reduced epidermal thickness, and inhibited inflammatory factor expression, suggesting a potential therapeutic effect. Further studies revealed that IDG promotes nuclear translocation of AhR and inhibits phosphorylation of NFκB p65, suggesting that its mechanism of action may involve the AhR/NFκB pathway. These findings indicate that IDG could be a promising drug candidate for the treatment of psoriasis. However, many issues need further correction.
1. Overall, this introduction provides a solid foundation for the research and successfully establishes the scientific rationale for the study. While mentioning IDG's anti-inflammatory effects, the introduction could benefit from specifying the specific mechanisms or pathways through which IDG exerts its anti-inflammatory actions. In addition, the transition from the paragraph about NFκB inhibitors to the paragraph about IDG could be improved with a smoother sentence that links the two concepts.
2. While the Materials & Methods section provides a good starting point, While the abstract mentions ethical approval, it lacks details about the housing conditions, monitoring for distress, and euthanasia criteria. The Materials & Methods should explicitly describe the measures taken to ensure animal welfare throughout the experiment. In addition, the software used for statistical analysis and image analysis should be explicitly mentioned.
3. The Materials & Methods section lacks detailed information on the catalog numbers of the reagents used. While the description of some reagents are included, the same level of detail should be provided for all other reagents and allowing other researchers to accurately replicate the study.
4. While you mention "significantly mitigated" and "notably ameliorated," in Figure 2 it would be more impactful to provide specific data about the reduction in PASI scores and the improvement in histological features. The text mentions a decrease in spleen index, but it doesn't explicitly explain why this is important or what it suggests about the therapeutic effects of IDG NE.
5. “Although the increase of IL-13 production in psoriasis-like lesions of the model group did not achieve statistical significance, it was also observed compared with the control group (Table 2).” The statement says IL-13 increase wasn't statistically significant but was also observed. This implies a contradiction. If something wasn't statistically significant, it means the observed difference could be due to random chance, and we can't confidently say it's a real effect.
6. The statement about HaCaT cells and LPS-stimulated models is well-established and doesn't require extensive citations in the results of “IDG inhibits inflammatory responses in LPS-stimulated keratinocytes”. The author can include that "To investigate the effects of indigo on psoriasis-related inflammation, we utilized an established LPS-stimulated HaCaT cell model. This model recapitulates key aspects of psoriasis pathogenesis, including the upregulation of inflammatory cytokines."
7. It's important to clarify whether the study focused specifically on the p65/p50 heterodimer or if other NF-κB dimers were also investigated in Figure 6.
8. While the Results mention the AhR/NF-κB pathway, it lacks a detailed explanation of the specific interaction between IDG, AhR, and NFκB pathways. A more comprehensive description of the proposed mechanism of action would enhance the scientific value of the Results.
9. The Discussion could be strengthened by providing a more detailed and specific explanation of how the findings support the overall conclusion. For example, the connection between the observed AhR activation and the inhibition of NF-κB could be more explicitly linked to the therapeutic effects of IDG NE.
10. While the Discussion mentions limitations, it could provide more detailed insights into these limitations and their implications for the interpretation of the findings.
11. The Discussion could be more impactful by directly addressing the clinical significance of the findings. For example, it could discuss how the study's results could translate into potential future therapies for psoriasis patients.
12. There are inconsistencies in spelling, for instance, "imiquimod" is spelled with and without a hyphen in the text. Consistent spelling throughout the manuscript is crucial for clarity.
13. Maintaining consistency in fonts and formatting across all figures and tables in your manuscript is essential for a professional and visually appealing presentation of your research. A consistent visual style enhances readability, clarity, and professionalism, making it easier for readers to understand and appreciate your findings. Using the same font for all figure titles, axis labels, legends, and table headings, maintaining consistent font sizes, and employing a standard format for captions all contribute to a more cohesive and polished manuscript.

Reviewer 2 ·

Basic reporting

In this study, the author explored the role of Indigo in psoriasis and demonstrated the AhR/NF-κB signaling pathway may be the underlying mechanism through an in vitro and in vivo study. The experimental design is rigorous, the content is detailed and the results supports the conclusion. However, there are still some deficiencies in details in the manuscript. Please revise it carefully according to the comments below.
1. In the method of abstract, “In this study…, and explored the mechanisms of these effects in vitro”, This is a conclusion description. Please rewrite this paragraph, including samples, animal, tissue processing, application dose and methods.
2. In the results of abstract, the description of the results should be which experiment showed which results. Please check and modify it carefully.
3. In the abstract, what is the significance of this article.
4. In the introduction, Pathogenesis the potential risk factors of psoriasis is what, Which gene mutations will affect the occurrence of psoriasis.
5. Which drugs are approved for the treatment of psoriasis. How is the treatment effect. How does their work theory. What are the therapeutic targets that have been used for psoriasis treatment.
6. The ary lhydrocarbon receptor (AhR) is a receptor, but in the line 58, the description of AhR is a ligand-activated cytoplasmic transcription factor. So, I'm confused, does the AhR have two functions, and what can be used as ligands for AhR.
7. NFκB is an inflammation regulator mediating inflammatory factors production, the AhR-NFκB signalling pathway plays a key role in the development of psoriasis. What is the relationship between AhR and NFκB. Does NFκB promote psoriasis by inhibiting the transcription of AhR,
8. What is the characteristic of AhR-NFκB signalling pathway. Please add more reports to the literature.
9. what are the anti-inflammatory and antioxidant mechanisms of Indigo.
10.Is this nanomaterial fabrication method common, how to evaluate its nanomaterial properties, and is there any literature supporting it.
11. The result section be modified emphatically. Some of the references cited in the current results are inappropriate and confusing. It is suggested to reorganize the results, mainly to explain the research purpose of each part and the specific results obtained by the author's experiment.
12. The scale number 100μm in Figure 2B is in the opposite direction and needs to be corrected.

Experimental design

Comments are merged into Basic reporting

Validity of the findings

Comments are merged into Basic reporting

Reviewer 3 ·

Basic reporting

This paper presents a study that investigates the therapeutic effects of Indigo (IDG) in psoriasis treatment through the AhR/NF-κB signaling pathway. The authors develope an IDG nanoemulsion and evaluate its effects on IMQ-induced psoriasis-like skin lesions in mice. They also study the mechanisms of action of IDG in vitro using HaCaT cells. The results demonstrate that IDG improves skin lesions, reduces inflammation, and activates AhR signaling.
The language of the manuscript will benefit from further proofreading in the native language so that the author's intention is clearer to the reader.
References are appropriate.
The figures should be improved in clarity. The white scale of pathology map is not very obvious, so it is recommended to adjust it to black or red.

Experimental design

The abstract section can be better described, the significance of the study needs to be clearly stated, and the description of the results should correspond to the method.
The study lacks detailed implementation details, such as the specific methods used for developing the IDG nanoemulsion and conducting the in vitro experiments. Providing more details would increase the reproducibility of the study.
The study could benefit from additional evaluation and ablation studies of the proposed method. For example, comparing the therapeutic effects of IDG with existing psoriasis treatments would strengthen the findings.

Validity of the findings

When using Luminex liquid phase chips to detect cytokines, it is recommended to provide the number of repetitions for each sample and the specific steps of the test to ensure the accuracy and reproducibility of the data.
The paper could benefit from a clearer discussion and comparison with existing literature on the topic. This would provide a better context for the study and highlight its novelty and contributions.
“Although the increase of IL-13 production in psoriasis-like lesions of the model group did not achieve statistical significance, it was also observed compared with the control group (Table 2).” Should be described more accurately.

Additional comments

The limitations of the study should be described.

---

## Round 0.2 · accepted · Accept

After careful consideration of the reviewers' comments and your revisions, I can confirm that you have adequately addressed all the points raised during the review process. Your efforts in responding to the reviewers' suggestions has improved the quality and clarity of your work. As such, I am happy to accept your manuscript for publication.

Reviewer 1 ·

Basic reporting

The article is basically clear and unambiguous. I have no other comment.

Experimental design

The investigation has been designed with a high technical standard.

Validity of the findings

The conclusions have been appropriately stated.

Reviewer 2 ·

Basic reporting

The author addresses my concerns well and has no further comments.

Experimental design

no comment

Validity of the findings

no comment

Reviewer 3 ·

Basic reporting

no comment.

Experimental design

no comment.

Validity of the findings

no comment.

Annotated reviews are not available for download in order to protect the identity of reviewers who chose to remain anonymous.